# Sedoanalgesia in the Debridement of Pediatric Burns in the Emergency Department: Is It Effective and Safe?

**DOI:** 10.3390/children10071137

**Published:** 2023-06-30

**Authors:** Carlos Delgado-Miguel, Miriam Miguel-Ferrero, Andrea Ezquerra, Mercedes Díaz, María De Ceano-Vivas, Juan Carlos López-Gutiérrez

**Affiliations:** 1Pediatric Burn Unit, Department of Pediatric Surgery, La Paz Children’s Hospital, 28046 Madrid, Spain; 2Institute for Health Research IdiPAZ, La Paz University Hospital, 28046 Madrid, Spain; 3Department of Pediatric Emergency, La Paz Children’s Hospital, 28046 Madrid, Spain

**Keywords:** sedoanalgesia, burns, children, emergency department, ketamine

## Abstract

Background: The routine use of sedoanalgesia has increased the number of potential minor surgical procedures that can be performed in the Emergency Department (ED) without requiring general anesthesia and, thus, hospital admission. Our aim is to analyze the effectiveness and safety of the use of sedoanalgesia in childhood burns treated in the ED. Methods: A retrospective study was conducted in burned children in whom burn debridement was performed under sedoanalgesia in the ED between 2017 and 2021 in a tertiary referral center for burns. We collected demographic variables, burn features and the type of sedoanalgesia performed in each case, including its effectiveness and associated adverse effects. Results: A total of 227 patients (118 males, 109 females) were included, with a median age of 25 months. In total, 99.2% of the burns were thermal (69.2% scald burns), with a mean total body surface area (TBSA) burned of 4%. The most commonly used drugs were intravenous ketamine (35.7%), intravenous ketamine + midazolam (15.4%), intranasal fentanyl + midazolam (14.1%) and intranasal fentanyl (10.6%). The effectiveness of sedoanalgesia was considered satisfactory in 95.2% of the cases, with an adverse effect rate of 7.5%, without severe adverse effects reported. Conclusions: The use of sedoanalgesia in the ED in the early treatment of childhood burns achieves high effectiveness and safety. It is postulated as a quality indicator; thus, it should be known by all pediatric healthcare practitioners.

## 1. Introduction

Anxiety, discomfort and anguish are common findings when a child is attended to in the Emergency Department. Many times, pain is the main complaint that gives rise to the need for medical attention. However, other times, pain and anxiety are the result of certain therapeutic and diagnostic techniques [1], and are frequently left undertreated due to the urgent nature of the procedures performed [2]. Recently, however, there has been a rising concern and commitment among ED medical staff to provide children with a safe and effective analgesia and sedation [3].

The use of sedoanalgesia has increased the number of minor surgical procedures that can be performed in the ED, such as suturing, the removal of foreign bodies or fracture management, without requiring general anesthesia and, thus, hospital admission. In 2017 we started to perform sedoanalgesia in the ED for early debridement of burns in children. Sedation induces a relaxed state and decreases consciousness. Analgesia relieves or diminishes the sensation to pain. Whereas numerous analgesic drugs can induce certain degree of sedation, most sedatives do not provide analgesia [4]. The combination of both effects, analgesia and sedation, enables pediatric surgeons to perform early debridement of minor burns in children in the ED without the need for hospital admission; however, to the best of our knowledge, no studies on this topic have been published to date. Our aim was to analyze our experience in the debridement of pediatric burns under sedoanalgesia in the ED, including the description of the patients treated, as well as the results obtained in terms of the effectiveness and safety of this procedure.

## 2. Materials and Methods

A retrospective study was conducted in patients with burns under 18 years of age in whom sedoanalgesia was performed for the debridement of burns in the ED between January 2017 and December 2021 in a tertiary referral center for burns. Patients in whom debridement was performed without sedoanalgesia and those in whom debridement was performed in the operating room were excluded.

We collected demographic data (gender, age, and weight), medical history (allergies, previous medical conditions, chronic treatment, American Society of Anesthesiologists Physical Status Classification System, (ASA score,) Mallampati class and recent respiratory infection), and burn features (etiology, depth, total body surface area burned and type of treatment performed), as well as the type of sedoanalgesia performed in each case (drug, route of administration, effectiveness and adverse effects).

Mallampati class was assessed according to the visibility of the pharyngeal structures and tongue, with the patient in a seated position and the mouth fully open. Grades I (visibility of soft palate, uvula and tonsillar pillars) and II (visibility of soft palate and uvula) were considered to be the absence of ventilation difficulties, while grades III (visible hard palate and base of uvula) and IV (only visible hard palate) were considered a risk of ventilation difficulties.

Recent respiratory infection was considered to be any respiratory infectious process affecting the upper and lower airways over the previous 3 weeks. All patients were monitored for heart rate and oxygen saturation using a pulse oximeter, and the room was supplied with oxygen masks, oxygen, and a positive-pressure ventilation bag in case these were required.

The effectiveness of sedoanalgesia was routinely recorded for all our patients as part of standard care in our institution by the emergency pediatrician supervising the procedure while the surgeon performed the debridement. The two aspects of sedoanalgesia were assessed: the degree of sedation and pain control. To evaluate sedation, the University of Michigan Sedation Scale (UMSS) was used [5], which gives a numerical score between 0 points (absence of sedation) and 4 points (complete sedation). These values were transformed into a three-category variable for statistical analysis: low effectiveness of sedation (0 points), adequate effectiveness of sedation (1–2 points), or high effectiveness of sedation (3–4 points); these categories were previously validated in previous studies [5].

The pain was evaluated during the debridement by the same pediatrician using the LLANTO scale, which is a reliable tool for Spanish speakers [6]. It consists of five items coded between 0 and 2 points, with a total score between 0 points (complete absence of pain) and 10 points (maximum pain). A score from 0–3 points is considered high pain management effectiveness; from 4 to 6 points, moderate effectiveness; and from 7 to 10 points, it is considered low pain management. These categories were derived from a systematic process previously published [6].

Prior to the procedure, written consent was obtained from the parents or legal guardians of the child for the debridement under sedoanalgesia. The protocol of the study obtained the approval of the institutional ethics committee and the hospital review board (PI-5410) and complied with the guidelines of the Declaration of Helsinki (1975). As this is a retrospective study, informed consent to participate was not required.

For statistical analysis, Microsoft Excel software (Redmond, WA, USA, 2010) and SPSS Statistic (Chicago, IL, USA, version 22) were used to collect and analyze data. The distributions of numerical variables were compared with Shapiro–Wilk and Kolmogorov–Smirnoff tests. Continuous variables not normally distributed were presented as medians and interquartile range (Q1–Q3), and continuous variables normally distributed were presented as means and standard deviation. Percentage (%) and frequency (n) were used to express categorical variables.

## 3. Results

A total of 227 patients (118 males, 109 females) were included, with a median age of 25 months (Q1–Q3: 15–61) and a mean weight of 19.8 ± 5.4 kg. Table 1 summarizes the patients’ medical history and potential anesthetic risk. A previous history of allergies was observed in 4.8% of the patients and medical conditions in 8.8%, with asthma and bronchospasm being the most frequent. When considering the ASA score, 91.6% of the patients were classified as ASA I, and 4.4% had Mallampati class III–IV. In these patients, a type of sedoanalgesia with a lower risk of respiratory depression, such as ketamine, was used.

Regarding burn features, thermal burns were the most frequent mechanism (99.2%), 69.2% of which were scalds, with water being the main etiological agent involved (51.0%) followed by oil (11.5%). The mean total body surface area (TBSA) burned was 4% (Q1–Q3: 2–6). In 124 patients (54.6%), superficial partial thickness burns were observed, and the remaining 103 (45.4%) presented deep partial thickness burns. The most frequent burn location was the upper extremities (30.4%), followed by the thorax–abdomen (23.8%) and the back (20.7%). Silver hydrocolloid dressing was the most frequently used treatment after debridement (69.1%), followed by nitrofurazone (11.9%) and silver sulfadiazine (12.5%). Table 2 lists the burn features and treatment modalities employed.

Concerning the kind of sedoanalgesia performed, intravenous ketamine (1.5 mg/kg) was the most frequently used drug (35.7%), followed by the combination of intravenous ketamine + midazolam (0.1 mg/kg) in 15.4% of the cases, combined intranasal fentanyl (1.5 mcg/kg) + intranasal midazolam (0.2 mg/kg) in 14.1% of the cases, and intranasal fentanyl alone in 10.6% of the cases. Type of sedoanalgesia, doses administered, sedation effectiveness and adverse effects are shown in Table 3.

The pain management effectiveness of each type of drug used in sedoanalgesia according to the LLANTO scale is described in Table 4. Intravenous ketamine shows the highest effectiveness, both alone and in combination with intravenous midazolam (85.3% and 85.8%, respectively), followed by intranasal ketamine, combined with midazolam (60%) or dexmedetomidine (61.5%). Intranasal fentanyl alone or in combination with midazolam is the least effective of the drugs used, being highly effective in only 12–30% of patients.

## 4. Discussion

Since we implemented this procedure in 2016 in our center, we have performed early burn debridement in 227 children without the need for hospital admission.

Today, the adequate prevention and treatment of pain and anxiety are considered a cornerstone of emergency care. All patients, children and adults, regardless of their past personal and medical history, have a right to receive safe and effective treatment for their pain and anxiety [7]. Many therapeutic and diagnostic procedures performed in the pediatric ED are painful or create anxiety in the patients and their parents [8]. Procedural sedoanalgesia has, therefore, become a most valuable and necessary tool in the ED. The first step to provide an efficient and safe sedoanalgesia to patients is to guarantee an appropriate training of all health care personnel involved in the procedure in order to minimize and promptly identify any potential side effect. Second, specific protocols must be developed to correctly identify which procedures would benefit from the administration of sedoanalgesia. In our center, any procedure that requires sedoanalgesia is carried out by a multidisciplinary unit composed of pediatricians, pediatric surgeons, and nurses [7]. Surgeons attend to patients with burns, and establish the need for sedoanalgesia for burn debridement. Pediatricians perform sedoanalgesia and monitor its effectiveness and safety.

In places where there is a scarcity of anesthesiologists and other medical staff frequently lack training in sedoanalgesia, many centers do not perform sedoanalgesia during painful procedures [9]. However, several case series have proven that sedoanalgesia can be administered safely and effectively by pediatricians who have received advanced life support training [10,11]. In our hospital, all ED pediatricians are trained in performing sedoanalgesia for painful procedures, and are an indispensable part of the multidisciplinary team. The treatment of burns in the ED under sedoanalgesia reduces the need for treatment in the operating room under general anesthesia, which would require the availability of anesthetists. It also reduces the rate of hospital admissions.

In our study, most of the patients were male, with a median age of 25 months. Consistently with previous studies, the majority of the children were in good health, had no abnormal findings on the physical examination, and were not on long-term treatments [12]. Sedoanalgesia can only be performed safely by non-anesthesiologists on patients with an ASA score of I or II. Although 8.8% of our patients had previous medical conditions (asthma and bronchospasm being the most frequent), none of them were classified as ASA III. The presence of a Mallampati class III–IV or a history of respiratory infection were not contraindications for sedoanalgesia, according to the clinical guidelines of the Spanish Society of Pediatric Emergencies, due to the low risk of associated complications [13]. In our center, we started to use sedoanalgesia in the ED for the debridement of burns in children a few years ago. Initially, we started treating only burns < 5% TBSA; however, due to the satisfactory outcomes obtained and the absence of major complications associated with this procedure, we currently perform debridement under sedoanalgesia in the ED on burns up to 10% of the TBSA [14].

The type of sedoanalgesia employed (pharmacological agent) was decided on an individual basis for each patient by the pediatric surgeon and the pediatrician, considering age, the TBSA burned and the presence of a peripheral venous catheter. In order to select the most appropriate analgesic or sedative drug for a patient, and to minimize both the risk of side effects or of underdosing for concern of adverse events, medical staff must be adequately trained in the pharmacodynamics, therapeutic effect, dosage, and side effects of the most frequent pharmacological agents employed [15]. Intravenous ketamine was the most frequently used drug in more than half of the patients, which was indicated in patients with burns >5% TBSA and in all those with a peripheral venous access catheter. Ketamine is a very effective and safe drug, extensively used in pediatric patients, whose effect starts within a minute, peaking at 5–10 min, and full recovery is achieved within 60–120 min [16]. It induces a cataleptic-like trance state via the dissociation of the limbic and thalamocortical systems. Thus, the central nervous system is “isolated” from external sensorial stimulation. As a result, protective airway reflexes, spontaneous breathing and cardiovascular dynamics are not compromised, whilst high degrees of sedation, amnesia and analgesia can be safely achieved [17]. Although ketamine has been traditionally used together with atropine to reduce hypersalivation, studies with a grade II level of evidence have shown that the degree of hypersalivation is similar whether or not anticholinergics are administered together with ketamine; for this reason, we do not use it in these patients [18]. It was highly effective in more than 85% of our patients when administered intravenously. We have recently started to use intranasal ketamine, associated with midazolam or dexmedetomidine, in patients under 3 years of age (in which cannulating a venous catheter can be challenging) with burns <5% of the TBSA. This route of administration is less effective (60%), but it is easier to administer as it does not require cannulation of a peripheral venous line. These drugs are administered via drops of the injectable solution into both nostrils with a 1 mL syringe for about 15 s; half of the dose is injected in each nostril. Using this approach, adequate sedation effects are quickly achieved, the placement of an intravenous line is not required, and, since it requires less collaboration from the patient, it shows benefits over the oral approach. The disadvantage is that it is not well-tolerated due to the nasal itching caused by the nasal pruritus, specifically produced by midazolam acid pH (pH 3.5) [19].

The overall sedation effectiveness was considered satisfactory in 95.2% of the cases. No cases of excessive sedation were recorded, probably due to the small number of cases treated with nitrous oxide, which has been the pharmacological agent most implicated in oversedation when administered at concentrations exceeding 70%; therefore, it should not exceed a concentration of 50%. [20]. This gas is especially useful in children older than 3–4 years who can collaborate in the procedure. The administration should be initiated three minutes before the start of the painful procedure painful procedure and be continued during the whole procedure. The administration should be temporarily discontinued if the patient is excessively sleepy [21].

In sedoanalgesia procedures, complications arise mainly from the depressive effects of the drugs used on circulation and the respiratory system. These complications usually occur 5 to 10 minutes after drug administration or immediately after the procedure. A total of 53 early adverse events were reported, with nystagmus being the most frequent of them (17.2%), followed by myoclonus (2.6%). Tachycardia was found in three patients (1.3%), and hypertension was found in three other patients (1.3%). All of these adverse reactions were resolved before the patient was discharged. The incidence of adverse events in other studies has a wide range, between 2 and 17%, possibly due to differences in the definition of complications [22,23]. Some authors do not consider nystagmus or myoclonus adverse reactions but physiological effects of ketamine. If we do not consider these reactions adverse events, the incidence in our study would be only 5.7%. The percentage of children who experienced vomiting was low (0.9%) and even lower than the percentage of cases described in other papers [10]. There is no conclusive proof yet of a correlation between a short fasting period and vomiting [24]. Many of the long-established recommendations in this regard have been extrapolated from the guidelines for procedures performed in the operating room under general anesthesia, which are not applicable to sedation procedures performed in the ED, both because of the type of drugs used and the degree of sedation achieved [25]. Regarding fasting times, the American Society of Anesthesiologists recommends precautions, although evidence is scarce; therefore, individual stratification should be carried out according to risk factors that depend on the patient’s characteristics and comorbidities, the technique or procedure to be performed, the degree of sedation, and mainly, depending on the urgency of the procedure [25]. In non-urgent procedures, following a fasting schedule is recommended: 2 h for clear liquids; 4 h for lactation liquids; 4 h for lactation; 6 h for non-fatty solids; and 8 h for full meals. However, in urgent procedures such as burn debridement, ingestion is not a contraindication and the appropriate precautions should be taken. The use of drugs with a lower risk of airway protective reflex depression, such as ketamine, or moderate sedation is recommended, if possible, to minimize the risks of sedation. In our case, given the unpredictability of this type of accident, and that the risk of postponing debridement is greater than the risk of a possible respiratory complication, the most controlled sedoanalgesia procedure is performed in order to debride the burn without pain as a matter of urgency. For these reasons, in our center, recent oral intake of food or liquids is not considered a contraindication for performing sedoanalgesia. In agreement with other published series, we did not observe any cases of aspiration of gastric contents [22].

When considering the discharge of a child who underwent burn debridement under analgesia, we must ensure that the child is going to be supervised by an adult, who is given instructions regarding potential late adverse events [10]. The patient should therefore remain in the emergency room with his guardians, monitored and supervised by competent personnel until complete recovery. The time until discharge will depend on the type of sedoanalgesia administered; however, in general, it is recommended to wait a minimum of 30 min provided that the ideal conditions are met, such as preserved airway and cardiovascular function with normal vital signs for age, adequate hydration, normal level of consciousness, and the patients should be alert, oriented, recognize their guardians and be able to speak, sit up and walk (if age appropriate) [26]. Once the patient’s condition has been assessed and discharge has been decided on, we must inform and give written instructions to family members or caregivers of the possibility of minor adverse effects and we must explain that the child should be under adult supervision for the following 24 h [27]. In our study, we found no late adverse events, probably because the patients remained under observation in the ED after the procedure, and were only discharged after being re-evaluated by pediatricians and when oral tolerance had been satisfactorily restarted.

The limitations of this study include its observational design, which was conducted retrospectively, and its exclusive focus on a single center. For these reasons, it would be interesting to perform a multicenter study with a larger number of patients. Further limitations are the lack of a control group, a cluster of children in which the painful burn debridement procedure was performed without sedoanalgesia; however, we believe this would be unfeasible due to ethical considerations. Moreover, due to the absence of previous similar studies, it is not possible to compare the results we have obtained. However, and despite its single center nature, we believe that the present review offers meaningful data since it has been carried out in the main referral hospital for pediatric burns in Spain. Nevertheless, a larger, multicenter study with more patients is warranted.

## 5. Conclusions

The utilization of sedoanalgesia as an early intervention for pediatric burn patients in the Emergency Department can be regarded as a viable and well-tolerated approach in ensuring effective pain management and safety in children. It could be helpful in achieving an effective debridement without pain and without serious adverse effects, which diminishes the need for treatment in the operating theatre under general anesthesia. It also enables outpatient treatment in selected patients, reducing the need for hospital admission. However, further prospective studies with a larger number of patients are still needed.

## Figures and Tables

**Table 1 children-10-01137-t001:** Patients’ medical history and potential anesthetic risk.

Allergies; n (%)	11 (4.8)
Medical conditions; n (%)	20 (8.8)
- Asthma	3 (1.3)
- Bronchospasm	6 (2.6)
- Laryngitis	1 (0.4)
- Inflammatory bowel disease	1 (0.4)
- Atopic dermatitis	2 (0.9)
- Celiac Disease	2 (0.9)
- Hypertension	2 (0.9)
- Autism spectrum disorder	1 (0.4)
- Patent foramen ovale	1 (0.4)
- Cleft palate	1 (0.4)
Chronic treatment; n (%)	9 (4)
ASA classification; n (%)	
● I	208 (91.6)
● II	19 (8.4)
● III	0
Mallampati class III–IV; n (%)	10 (4.4)
Recent respiratory infection; n (%)	31 (13.7)

**Table 2 children-10-01137-t002:** Burn characteristics and the type of treatment performed.

Type of burn, n (%)	
● Thermal	225 (99.2)
○ Scalding	157 (69.2)
● Water	80 (51.0)
● Oil	18 (11.5)
● Soup	17 (10.8)
● Puree	10 (6.4)
● Tea	12 (7.6)
● Coffee	15 (9.6)
● Caramel	5 (3.2)
○ Others	68 (30.8)
● Lighter	7 (10.3)
● Iron	10 (14.7)
● Firecracker	5 (7.4)
● Oven	8 (11.8)
● Radiator	8 (11.8)
● Fireplace	6 (8.8)
● Ceramic hob	7 (10.3)
● Candle flame	3 (4.4)
● Candle wax	5 (7.4)
● Light bulb	3 (4.4)
● Charcoal	2 (2.9)
● Slide	1 (1.5)
● Solar	3 (4.4)
● Chemical (sulfuric acid)	1 (0.4)
● Electrical (domestic socket)	1 (0.4)
Depth; n (%)	
● Superficial partial thickness	124 (54.6)
● Deep partial thickness	103 (45.4)
TBSA * burned (%); median (Q1–Q3)	4 (2–6)
Burn location; n (%)	
● Upper limbs	69 (30.4)
● Thorax-abdomen	54 (23.8)
● Back	47 (20.7)
● Lower limbs	43 (18.9)
● Head and neck	14 (6.2)
Type of treatment; n (%)	
● Silver hydrocolloid (Aquacel Ag^®^)	157 (69.1)
● Nitrofurazone (Furacin^®^)	27 (11.9)
● Silver Sulfadiazine (Silvederma^®^)	26 (11.5)
● Microporous membrane (Suprathel^®^)	17 (7.5)

* TBSA, total body surface area.

**Table 3 children-10-01137-t003:** Type of sedoanalgesia, doses administered, sedation effectiveness and adverse events reported.

Type of Sedoanalgesia used, n (%)	
● IN Fentanyl (1.5 mcg/kg)	24 (10.6)
● IN Fentanyl (1.5 mcg/kg) + nitrous oxide (O_2_ + N_2_O at 50%)	10 (4.4)
● IN Fentanyl (1.5 mcg/kg) + IN Midazolam (0.2 mg/kg)	325. (14.1)
● IV Ketamine (1.5 mg/kg)	81 (35.7)
● IV Ketamine (1.5 mg/kg) + IV Midazolam (0.1 mg/kg)	35 (15.4)
● IN Ketamine (4 mg/kg) + IN Midazolam (0.2 mg/kg)	27 (11.9)
● IN Ketamine (4 mg/kg) + IN Dexmedetomidine (2 mcg/kg)	18 (7.9)
Sedation effectiveness according to UMSS scale, n (%)	
● High (3–4 points)	162 (71.3)
● Adequate (1–2 points)	56 (24.7)
● Low (0 points)	9 (4)
Adverse events, n (%)	53 (23.3)
Type of adverse event, n (%)	
● Nystagmus	39 (17.2)
● Myoclonus	6 (2.6)
● Exanthema	2 (1.4%)
● Tremor	3 (2.1%)
● Hypertension	3 (1.3)
● Tachycardia	3 (1.3)
● Vomiting	2 (0.9)

IN, intranasal; IV, intravenous.

**Table 4 children-10-01137-t004:** Pain management effectiveness of each type of drug used in sedoanalgesia according to LLANTO scale.

	High (0–3 Points)	Moderate (4–6 Points)	Low (7–10 Points)
IN Fentanyl	12.5%	75%	12.5%
IN Fentanyl + nitrous oxide	100%	-	-
IN Fentanyl + IN Midazolam	25%	75%	-
IV Ketamine	85.3%	14.7%	-
IV Ketamine + IV Midazolam	85.8%	12.1%	2.1%
IN Ketamine + IN Midazolam	60%	40%	-
IN Ketamine + IN Dexmedetomidine	61.5%	30.8%	7.7%

## Data Availability

The data generated and analysed in this study will be made available upon reasonable request to the authors.

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
