# Peer review of "Sedoanalgesia in the Debridement of Pediatric Burns in the Emergency Department: Is It Effective and Safe?"

_children, 2023, doi:10.3390/children10071137_

Round 1
Reviewer 1 Report
1. I would reword your aims statements for consistency, as this will impact how your methods are judged. Was your aim to simply describe your experience using sedoanalgesia for early debridement of pediatric burns in the ED, or was it to analyze effectiveness and safety? The latter would require a higher standard to be met.
2. If your aim was to truly comment on safety, your sample size is too small to comment as strongly as you have on safety. This is also particularly due to the fact that the majority of patients were treated with ketamine, which has a good safety profile and associated with very low incidence of adverse events, especially low incidence of serious adverse events. I would temper your language regarding safety accordingly.
3. Your assessment of effectiveness is confusing. In your methods, you make mention of both LLANTO and UMSS. However, I do not see any report of the LLANTO scores. I also think that you need to be more specific in stating that your definition of effectiveness, as it appears in your study, is effectiveness in achieving a specific DEPTH of sedation. Effectiveness is a very general and non-specific term in sedation research and can be used to mean a multitude of different things - effective in producing high quality of sedation? quick onset of action? anxiolysis? and so forth. You need to be more explicit in stating that when you refer to effectiveness, you are only commenting on the medications' ability to achieve what you have defined as an adequate depth of sedation - nothing more.
4. Do you have information about location of burns? In addition to the data you have already provided, this would enhance the reader's understanding of patients suitable for debridement in the ED setting using the methods you have described.
5. Was written consent obtained for the procedure or to collect data for the study? It is unclear as written.
6. Was LLANTO and UMSS collected as part of standard practice in all debridements you reported? Not all institutions routinely collect this data during their sedations or procedures. If not, I would be concerned about missing data in the context of a retrospective study.
Author Response
Reviewer 1
- I would reword your aims statements for consistency, as this will impact how your methods are judged. Was your aim to simply describe your experience using sedoanalgesia for early debridement of pediatric burns in the ED, or was it to analyze effectiveness and safety? The latter would require a higher standard to be met.
Thank you very much for your comment. The aim of this study is to analyse our experience in the debridement of paediatric burns under sedoanalgesia in the Emergency Department, including the description of our patients treated, as well as the results obtained in terms of effectiveness and safety of this procedure. We have reformulated the objective in the Introduction section following your recommendation.
- If your aim was to truly comment on safety, your sample size is too small to comment as strongly as you have on safety. This is also particularly due to the fact that the majority of patients were treated with ketamine, which has a good safety profile and associated with very low incidence of adverse events, especially low incidence of serious adverse events. I would temper your language regarding safety accordingly.
Thank you for your comment, we agree that it is a small sample size study to make such strong conclusions about safety. Indeed, ketamine was the main drug used, mainly because of its low rate of adverse effects. Following your recommendation, we have modified the conclusions of the paper, considering that sedoanalgesia in burn debridement "may be" a procedure with a low risk of adverse effects, although prospective studies with a larger number of patients are needed.
- Your assessment of effectiveness is confusing. In your methods, you make mention of both LLANTO and UMSS. However, I do not see any report of the LLANTO scores. I also think that you need to be more specific in stating that your definition of effectiveness, as it appears in your study, is effectiveness in achieving a specific DEPTH of sedation. Effectiveness is a very general and non-specific term in sedation research and can be used to mean a multitude of different things - effective in producing high quality of sedation? quick onset of action? anxiolysis? and so forth. You need to be more explicit in stating that when you refer to effectiveness, you are only commenting on the medications' ability to achieve what you have defined as an adequate depth of sedation - nothing more.
We agree with your assessment, as on reading the article again, it is not clear how the effectiveness of the procedure is measured. The effectiveness of sedoanalgesia was measured using two scales: one to measure sedation status (University of Michigan Sedation Scale, UMSS), and another to measure acute pain (the LLANTO scale), which is a reliable tool for Spanish speakers validated from the CHEOPS scale.
Both scales give a numerical score, 0-4 points for the UMSS scale, and 0-10 points for the LLANTO scale. These numerical data were transformed into qualitative variables to assess sedation (high, adequate, low) as well as pain (high, moderate, low). We have added these clarifications in the Methods section, as well as in the Results section and in Table 3 and Table 4 of the manuscript.
- Do you have information about location of burns? In addition to the data you have already provided, this would enhance the reader's understanding of patients suitable for debridement in the ED setting using the methods you have described.
Effectively, the location of the burns is another of the data routinely recorded on the sedoanalgesia prescription sheet, together with the body surface area affected. The main location of the burns was the upper extremities, followed by the thorax-abdomen, back and lower extremities. We have added these data in the Results section as well as in Table 3 of the manuscript.
- Was written consent obtained for the procedure or to collect data for the study? It is unclear as written.
Written consent obtained from the parents for the procedure, prior to debridement under sedoanalgesia. As this is a retrospective study, informed consent was not required. We have added this clarification in the Methods section.
- Was LLANTO and UMSS collected as part of standard practice in all debridements you reported? Not all institutions routinely collect this data during their sedations or procedures. If not, I would be concerned about missing data in the context of a retrospective study.
Thank you very much for your comment. Indeed, the LLANTO scale score is collected in a standardised manner during all sedoanalgesia procedures in our Emergency Department. To facilitate this collection, there is a table with the variables measured by the scale (cry, attitude, breathing, postural tone and facial observation) on the sedoanalgesia sheet itself, which is filled in for each patient with clinical and pharmacological data. This has allowed us to carry out retrospective studies without losing important data such as the effectiveness of sedoanalgesia.
Reviewer 2 Report
Dear Authors
I have read with interest the article „Sedoanalgesia In The Debridement Of Pediatric Burns In The Emergency Department: Is It Effective And Safe?“
These are my comments:
Major:
1. 1. It is unclear how exactly the effectiveness of sedoanalgesia was assessed. The authors describe the assessment of the adequacy of sedation (insufficient, adequate and high) and pain scores. However in the results section efficacy is described as low, adequate and high. On what parameters was it based? Were pain scores included into the effectiveness assessment? Or only sedation scores were used? This should be clarified in the methods section, as the efficacy or effectiveness is the main outcome measure.
2. 2. In methods section pain assessment is described by the use of LLANTO scale, however, no pain scores are provided in the result section. This should be corrected.
3. 3. Please include in the methods or results section real fasting times and fasting guidelines in your department before wound dressing under analgosedation. Please discuss this topic more widely in the discussion section. As this topic is very important and interesting for the reader.
4. 4. Table 1. Risk of difficult airway was observed in 10 patients. This part should be elaborated: please, include diagnoses or symptoms indicating difficult airway in these 10 patients. Please also describe in detail what kind of analgosedation was used in these patients and did any complications occur in this patient group.
5. 5. Table 1. Recent respiratory infection – please explain what time spam was considered as „recent“. Please discuss what risks can be associated with recent respiratory infections and were there any complications in these patients.
6. 6. Table 4. Please include the the dose range for every sedative agent used by every route. Please provide wider discussion on the agents used, comparing the doses or referring to safety, efficacy and pharmacokinetic data where appropriate (with previous studies).
Author Response
Reviewer 2
- It is unclear how exactly the effectiveness of sedoanalgesia was assessed. The authors describe the assessment of the adequacy of sedation (insufficient, adequate and high) and pain scores. However in the results section efficacy is described as low, adequate and high. On what parameters was it based? Were pain scores included into the effectiveness assessment? Or only sedation scores were used? This should be clarified in the methods section, as the efficacy or effectiveness is the main outcome measure.
The effectiveness of sedoanalgesia was measured using two scales: one to measure sedation status (University of Michigan Sedation Scale, UMSS), and another to measure acute pain (the LLANTO scale), which is a reliable tool for Spanish speakers validated from the CHEOPS scale.
Both scales give a numerical score (0-4 points for the UMSS scale) and 0-10 points for the LLANTO scale. These numerical data were transformed into qualitative variables to assess sedation (high, adequate, low) as well as pain (high, moderate, low). We have added these clarifications in the Methods section, as well as in the Results section and in Table 3 of the manuscript.
- In methods section pain assessment is described by the use of LLANTO scale, however, no pain scores are provided in the result section. This should be corrected.
LLANTO scale has 5 variables coded between 0 and 2 points, so the total score is between 0 points (total absence of pain) and 10 points (maximum pain). From 0-3 points is considered high pain management effectiveness, from 4-6 points moderate effectiveness and from 7-10 points low pain management. We have added this clarification both in the Methods section and in Table 3 where the effectiveness of the sedoanalgesia performed is shown.
- Please include in the methods or results section real fasting times and fasting guidelines in your department before wound dressing under analgosedation. Please discuss this topic more widely in the discussion section. As this topic is very important and interesting for the reader.
Regarding fasting times, they should be assessed on an individual basis. The American Society of Anesthesiologists recommends precautions, although evidence is scarce. Individual stratification should be carried out according to risk factors that depend on the patient's characteristics and comorbidities, the technique or procedure to be performed, the degree of sedation, and mainly, depending on the urgency of the procedure. In non-urgent procedure it is recommended to follow fasting schedule: 2 hours for clear liquids, 4 hours for lactation liquids, 4 hours for lactation, 6 hours for non-fatty for non-fat solids and 8 hours for full meals. However, in urgent procedure such as burn debridement, ingestion is not a contraindication and the appropriate precautions should be taken. It is suggested that the possibility of using drugs with a lower risk of airway protective reflex depression, such as ketamine, or moderate sedation to minimise risks sedation if possible. In our case, given the unpredictability of this type of accident, and that the risk of postponing debridement is greater than the risk of a possible respiratory complication, the most controlled sedoanalgesia procedure is performed in order to debride the burn without pain as a matter of urgency. We have added this clarification in the Discussion section.
- Table 1. Risk of difficult airway was observed in 10 patients. This part should be elaborated: please, include diagnoses or symptoms indicating difficult airway in these 10 patients. Please also describe in detail what kind of analgosedation was used in these patients and did any complications occur in this patient group.
The risk of difficult airway was considered according to the Mallampati classification, where the visibility of pharyngeal structures and tongue is assessed, with the patient in a seated position and the mouth fully open. Grades I (visibility of soft palate, uvula and tonsillar pillars) and II (visibility of soft palate and uvula) were considered as absence of difficult airway, while grades III (visible hard palate and base of uvula) and IV (only visible hard palate) were considered as risk of difficult airway. In these patients, a type of sedoanalgesia with a lower risk of respiratory depression such as ketamine was used. We have added these clarifications in both the Methods and Discussion sections.
- Table 1. Recent respiratory infection – please explain what time spam was considered as “recent“. Please discuss what risks can be associated with recent respiratory infections and were there any complications in these patients.
Recent respiratory infection was considered as any respiratory infectious process affecting the upper and lower airways during the last 3 weeks. This is the time in which bronchial hyperresponsiveness persists after a respiratory infection, and in scheduled procedures, this is the time in which it is recommended to postpone anaesthesia for this reason. However, as with fasting time, in emergency procedures, such as burn debridement, there is no possibility of delaying it for 3 weeks, so drugs with a low risk of bronchial hyperresponsiveness, such as ketamine, are used. We have added this clarification in the Methods and Discussion section.
- Table 4. Please include the the dose range for every sedative agent used by every route. Please provide wider discussion on the agents used, comparing the doses or referring to safety, efficacy and pharmacokinetic data where appropriate (with previous studies).
We have added the doses for each sedative agent and for each route in Table 3. In addition, in the Discussion we have added a paragraph commenting on the advantages and disadvantages of each drug and each route.
Round 2
Reviewer 1 Report
Thank you for addressing all the concerns raised. There are only two points of clarification I would add:
1. State explicitly in your manuscript that the UMSS and LLANTO scores are routinely recorded for all your patients as part of standard care for your institution (by the emergency pediatrician supervising the procedure - this latter part, I recognize you have already stated).
2. For the statements attributing qualitative descriptors to scores for UMSS and LLANTO scale (e.g. insufficient sedation = 0 points, adequate sedation = 1-2 points; 0-3 points is considered high management pain management effectiveness, 4-6 points moderate effectiveness, etc.) state whether these categories were created by you (the investigators) or if they were derived from a systematic process. If the latter, please cite the references. If you created them, simply state something to the effect of, "These categorizations were created by our study team." Include this statement both after describing the categories for UMSS and after describing the categories for LLANTO.
Author Response
Thank you for addressing all the concerns raised. There are only two points of clarification I would add:
- State explicitly in your manuscript that the UMSS and LLANTO scores are routinely recorded for all your patients as part of standard care for your institution (by the emergency pediatrician supervising the procedure - this latter part, I recognize you have already stated).
We have added this clarification, following the reviewer's recommendation, in the Methods section. (“The effectiveness of sedoanalgesia was routinely recorded for all our patients as part of standard care in our institution by the emergency pediatrician supervising the procedure while the surgeon performed the debridement”).
- For the statements attributing qualitative descriptors to scores for UMSS and LLANTO scale (e.g. insufficient sedation = 0 points, adequate sedation = 1-2 points; 0-3 points is considered high management pain management effectiveness, 4-6 points moderate effectiveness, etc.) state whether these categories were created by you (the investigators) or if they were derived from a systematic process. If the latter, please cite the references. If you created them, simply state something to the effect of, "These categorizations were created by our study team." Include this statement both after describing the categories for UMSS and after describing the categories for LLANTO.
Indeed, these categories were derived from a systematic process previously published by other authors, which we have added in the references.
Reviewer 2 Report
Dear Authors,
The revised version of an article has improved. These are further changes which should be made (line numbers are according the corrected version).
Methodology:
1. Risk of difficult airway – Please change „difficult airway‘‘ To „Mallampati class“ everywhre throughaut the text (including Tables) as difficult airway is quite different assumption, and Mallampati class is only one of the potential symptoms of difficult airway. Describe Mallampati class in the new paragraph.
2. Recent respiratory infection – describe in the new paragraph.
3. Line 79: „validated from the CHEOPS scale” this part of the sentence should be excluded as it is not true or meaning is obscure due to inappropriate English.
Consistency of terminology:
4. Please change „Insufficient sedation“ to „Low effectiveness of sedation“ , „adequate sedation“ to „adequate effectiveness of sedation“, „high sedation“ to „High effectiveness of sedation“
Results:
5. consistency of terminology in Table 4 (effectiveness of sedoanalgesia): „adequate“ please change to „moderate“ (as was provided in the Methods section)
Discussion:
6. please exclude the first unnecesary sentence giving no additional information „To the best of our knowledge, this is the first study to describe..” Such statements should be avoided in scientific literature. Start directly with the the following info (Since…).
7. Line 160: “in developing countries..” should be excluded as the expression may look unacceptable to some readers and statement is not based by the title of the citing reference.
8. Line 217 – Nitrous oxide is never recommended to be administered in concentration exceeding 70%. Please correct.
9. Lines 224-226 make no sense and do not reflect reality. Please exclude.
10. Line 77: reference [1322] - please correct
Language
Chief complain to main complain
Lines 211-212 needs correction
Lines 220-223 makes no sense – please correct.
Line 268- 272 should be corrected
Author Response
Dear Authors,
The revised version of an article has improved. These are further changes which should be made (line numbers are according the corrected version).
Methodology:
- Risk of difficult airway – Please change „difficult airway‘‘ To „Mallampati class“ everywhre throughaut the text (including Tables) as difficult airway is quite different assumption, and Mallampati class is only one of the potential symptoms of difficult airway. Describe Mallampati class in the new paragraph.
We have changed the term "difficult airway'' to "Mallampati class" everywhre throughaut the text, including Tables, and separated it into a new paragraph in the Methods section.
- Recent respiratory infection – describe in the new paragraph.
Following the reviewer's recommendations, we have added a new paragraph to describe recent respiratory infection in the Methods section.
- Line 79: „validated from the CHEOPS scale” this part of the sentence should be excluded as it is not true or meaning is obscure due to inappropriate English.
We have removed this part of the sentence, according to the reviewer's recommendation.
- Please change „Insufficient sedation“ to „Low effectiveness of sedation“ , „adequate sedation“ to „adequate effectiveness of sedation“, „high sedation“ to „High effectiveness of sedation“
We have modified these terms in the Methods section.
Results:
- Consistency of terminology in Table 4 (effectiveness of sedoanalgesia): „adequate“ please change to „moderate“ (as was provided in the Methods section)
We have modified this terminology, according to the reviewer's recommendation.
Discussion:
- Please exclude the first unnecesary sentence giving no additional information „To the best of our knowledge, this is the first study to describe..” Such statements should be avoided in scientific literature. Start directly with the the following info (Since…).
Following the reviewer's recommendation, we have removed this part of the sentence,.
- Line 160: “in developing countries..” should be excluded as the expression may look unacceptable to some readers and statement is not based by the title of the citing reference.
We have removed this part of the sentence, according to the reviewer's recommendation.
- Line 217 – Nitrous oxide is never recommended to be administered in concentration exceeding 70%. Please correct.
Indeed, administration above 70% is never recommended because of the high risk of excessive sedation. We have clarified this concept in the manuscript.
- Lines 224-226 make no sense and do not reflect reality. Please exclude.
We have removed this sentence, according to the reviewer's recommendation.
- Line 77: reference [1322] - please correct
We have modified the typographical error.
Language
Chief complain to main complain.
We have already modified this sentence.
Lines 211-212 needs correction.
We have already modified this sentence.
Lines 220-223 makes no sense – please correct.
We have already modified this sentence.
Line 268- 272 should be corrected.
We have already modified this sentence.